# Correction of Sea Surface Biases in the NEMO Ocean General Circulation Model Using Neural Networks

Andrea Storto[1,2], Sergey Frolov[3], Laura Slivinski[3], Chunxue Yang[1,2]

[1]National Research Council of Italy (CNR), Institute of Marine Sciences (ISMAR), Rome, Italy.
[2]National Research Center for High Performance Computing, Big Data and Quantum Computing (ICSC), Italy
[3]National Oceanic and Atmospheric Administration (NOAA), Physical Sciences Laboratory (PSL), Boulder, CO, United States

*Correspondence to*: Andrea Storto (andrea.storto@cnr.it)

**Abstract.** The atmospheric forcing and the heat exchanges between the ocean and the atmosphere represent one of the major sources of uncertainty for numerical ocean reconstructions and predictions, together with inaccuracies in vertical mixing and
solar radiation penetration. Air-sea heat fluxes may suffer from inaccuracies in meteorological fields, sea surface variables, and bulk formulations, which have a strongly non-linear dependence on the ocean state. Here, state-dependent errors of the heat fluxes are learned by artificial neural networks (ANN) from a dataset of heat flux correction terms, derived in turn from previous sea surface temperature nudging experiments. The pre-trained model predictors include stationary fields, atmospheric forcing data, ocean state, and stratification indices. Variable importance scores emphasize the dependence of the air-sea heat
flux errors on the wind forcing. The pre-trained model of heat flux correction is then used to adaptively correct the fluxes online, in a series of global ocean experiments performed with the NEMO version 4 (Nucleus for European Modelling of the Ocean) ocean general circulation model, augmented with ANN inference capabilities in Fortran90. Results indicate the positive impact of the correction procedure, beyond the training period, e.g., in independent observation-poor and -rich periods, leading to the same dynamic and subsurface signature as in nudging experiments. Prediction experiments also indicate the method's
potential for operational forecast applications. The method may also be adopted in coupled long-term reanalyses, long-range predictions, and projections.

**Short summary.** Inaccuracies in air-sea heat fluxes severely degrade the accuracy of ocean numerical simulations. Here, we use artificial neural networks to correct the air-sea heat fluxes as a function of oceanic and atmospheric state predictors. The correction successfully improves surface and subsurface ocean temperatures beyond the training period and in prediction
experiments.

## 1 Introduction

The ocean and the atmosphere interact by exchanging momentum, heat, and freshwater. These interactions drive ocean circulation and ventilation (e.g., Marzocchi et al., 2021), its energy and water budgets, which are crucial to understanding the ocean's role in Earth's climate and its variability over a wide range of spatial and temporal scales (e.g., Roberts et al., 2016;

Small et al., 2019). Unfortunately, direct measurements of these fluxes are only available in limited buoy locations, making their global and precise estimate a challenging problem (Cronin et al., 2019). Typically, air-sea fluxes are estimated using bulk flux parameterizations, which rely on near-surface meteorological variables, obtained from numerical weather prediction systems or atmospheric reanalyses (e.g., Yu, 2019). Bulk formulations are strongly non-linear, and there are significant uncertainties in these parameterization-based flux estimates (e.g., Huber and Zanna, 2017); when averaged over ocean basins,

heat fluxes may result in considerable imbalances (see, e.g., Kato et al., 2013; Storto et al., 2016; Valdivieso et al., 2017). Inaccuracies in ocean model vertical mixing and solar radiation penetration schemes interplay with the inaccuracies in the air-sea fluxes, and may amplify the sea surface errors (e.g., Deppenmeier et al., 2020, Jia et al., 2021, Richards et al., 2009).

For both retrospective ocean simulations (e.g., OMIP, Ocean Model Intercomparison Project, Griffies et al., 2016), long-term reanalyses (Storto et al., 2021) and coupled model simulations (e.g., CMIP, Coupled Model Intercomparison Project, Small et

al., 2019), systematic errors at the sea surface affect ocean heat redistribution (convection, stratification, and large-scale circulation), potentially compromising climate change signals (Storto et al., 2016; Carton et al., 2018). Errors at the air-sea interface thus remain among the most critical sources of uncertainty for many numerical ocean applications, including climate monitoring (e.g., Hakuba et al., 2024) and operational forecasting (e.g., Lewis et al., 2019; Lin et al., 2023; Ohishi et al., 2024). Attempts to empirically correct errors in the fluxes have generally developed along two directions: i) bias-correction

methodologies applied directly to ocean variables, i.e. correcting the effects of the air-sea heat flux systematic errors, see e.g. Balmaseda et al. (2007); ii) calibrating atmospheric reanalyses through comparison with observed climatology (Large and Yeager, 2009; Brodeau et al., 2010; Tsujino et al., 2018). Both strategies have their merits and weaknesses; bias-correcting ocean variables requires an adequate and dense ocean observing network, namely relying on the Argo float network limited to the period from ~2005 onwards, and cannot be used for attributing ocean model errors to specific processes; on the other hand,

calibrating atmospheric reanalyses can mitigate errors in the atmospheric forcing, but not in the bulk formula approximations, and therefore is only partially able to improve air-sea heat fluxes. Stochastic approaches can also, to some limited extent, improve the estimation of air-sea heat fluxes through rectification of the mean ocean state (Agarwal et al., 2023; Storto and Yang, 2023).

In this work, we use a state-of-the-science ocean general circulation model to showcase the idea of finding a predictor-

correction empirical relationship, formulated in terms of neural networks, to correct the non-solar component of the air-sea heat fluxes and reduce the sea surface temperature biases. As neural networks have been proven to be universal approximators of any function (Hornik et al., 1989), they represent an obvious and flexible choice to model non-linear relationships between the atmospheric and oceanic states and the heat flux errors. Indeed, previous work (Bonavita and Laloyaux, 2020; Chen et al., 2022; Chapman and Berner, 2024) has shown their ability to infer systematic errors in atmospheric models. The use of data

assimilation increments was also demonstrated to be a robust strategy to learn such errors, with both theoretical (e.g., Mitchell and Carrassi, 2015) and practical (Farchi et al., 2021; Chapman and Berner, 2024) arguments. The relationship is learned offline from present-day ocean model simulations that exploit the availability of space-borne sea surface temperature to

estimate a corrective heat flux term. The correction is then tested online in ocean model simulations, for periods beyond the learned (training) one.

The article's structure is as follows: after this Introduction, Section 2 describes the modeling system, the neural network setup, the relevant datasets, and the experimental setup. Section 3 summarizes the results of the reconstruction of the corrective heat flux terms and the online correction experiments, while Section 4 discusses and concludes.

## 2 Materials and Methods

### 2.1 The NEMO model and the nudging scheme

In this work, we use the NEMO ocean model (version 4.0.7, Madec et al., 2017) including the sea-ice dynamic and thermodynamic model SI[3]. NEMO is implemented on the ORCA1 grid (at 1° of horizontal resolution with refinement in the Tropics), with 75 vertical depth levels and partial steps (Barnier et al., 2006). We use the same model configuration as in the CIGAR reanalysis (Storto and Yang, 2024), briefly recalled here. The surface boundary conditions are calculated through the CORE bulk formulas (Large and Yeager, 2009) implemented in the AEROBULK package (Brodeau et al., 2016), using

meteorological variables extracted from the ECMWF ERA5 atmospheric reanalysis (Hersbach et al., 2020). The river discharge from land is provided by the JMA JRA-55-do reanalysis (Tsujino et al., 2018). The model setup includes i) a 3-band RGB scheme for the net shortwave radiation, with extinction coefficients that depend on a monthly climatology of chlorophyll; ii) the TKE scheme for the vertical mixing (Gaspar et al., 1990); iii) a Laplacian operator and a bi-Laplacian operator for tracers and momentum, respectively.

In the NEMO model, the air-sea heat flux can be optionally corrected with a nudging scheme (see e.g., Storto et al., 2016b). In practice, the net heat flux is decomposed into a penetrative (solar) component and a non-penetrative (non-solar) component. The non-solar component, which includes latent, sensible, and net longwave heat flux, can be corrected as:

$$Q'_{ns} = Q_{ns} + Q_{rp} = Q_{ns} + \kappa \left( SST_o - SST \right) \tag{1}$$

where the misfit between the observed ($SST_o$) and modeled ($SST$) sea surface temperature, multiplied by the nudging

coefficient (or strength) $\kappa$, represents the corrective flux $Q_{rp}$ added to the uncorrected non-solar flux. SST nudging is still a popular assimilation methodology for many climate-scale applications, where the use of gap-filled SST data ensures temporal consistency of the simulated ocean state compared to the direct assimilation of SST measurements (see, e.g., Yang et al., 2017). A 2000-2020 experiment (referred to as REF) with nudging to the SST data from the UKMO HadISST dataset (Rayner et al., 2003) was conducted, with a nudging coefficient equal to 100 W m$^{-2}$ K$^{-1}$, which roughly corresponds to a 20-day relaxation

time scale for a 50 m deep mixed layer. Note that nudging coefficients may be related to error characteristics and set up in a statistically optimal way (e.g., Zou et al., 1992; Vidard et al., 2003), although here, for the sake of simplicity, the nudging coefficient $\kappa$ is spatially and temporally constant. Additionally, preliminary experiments tested the use of alternative SST

datasets, for instance, the NOAA DOISST v2.1 (Huang et al., 2021), but those using HadISST provided the best results, and are the only ones considered in the remainder of the article.

Correcting air-sea heat fluxes effectively accounts for multiple sources of bias in the modelled sea surface temperature (SST), including potential errors in vertical mixing and other oceanic processes. Since the bias is assessed against observations without possible attribution to a specific error source, the method serves as a general SST bias correction strategy. Additionally, the correction is applied to the air-sea heat flux rather than directly modifying the SST tendency. Direct SST tendency corrections are generally unsatisfactory, as they require arbitrary assumptions about vertical propagation - such as confinement within the

mixed layer - or risk being nullified by air-sea interactions (see, e.g., Waters et al., 2015; Storto and Oddo, 2019). Adjusting air-sea heat fluxes is therefore a customary and physically consistent practice in ocean general circulation models; similar approaches are indeed used also by state estimation systems, such as ECCO4 (Forget et al., 2015), which employs observations to correct heat flux components.

## 2.2 Artificial Neural Networks

The artificial neural network (ANN) employs a feed-forward architecture to infer the corrective flux $Q_{rp}$ using several predictors. The gridded predictors (ocean model fields) are unrolled to form independent columns (gridpoint-wise data) and the geographical information is retained through the addition of longitude and latitude as predictors. This approach was referred to as column neural networks (e.g., Bonavita and Laloyaux, 2020). In the ANN, $Q_{rp}$ will not depend any longer on SST observations but on several input predictors, representative of the atmospheric and oceanic states, and detailed below.

We grouped the predictors into several categories, listed in Table 1, to represent different sources of errors: i) stationary errors (location and day of the month); ii) surface temperature and its diurnal cycle; iii) heat flux components; iv) atmospheric wind forcing; v) surface salinity and the freshwater components; vi) ocean stratification and its diurnal cycle. Within the ANN training, the input variables are taken as daily means from the REF experiment (with SST nudging enabled), except the variables referring to diurnal amplitudes (defined as the maximum value minus the minimum value, at hourly frequency, within

each day). The output fields used for training the ANNs are the $Q_{rp}$ fields from the REF experiment, taken as the average between the same day as the predictors and the following day, assuming it is nominally valid at the end of each daily window (midnight UTC).

Over sea-ice-covered areas, the heat flux corrections vanish, due to the use of a sea-ice-based weighting function - that zeroes the correction for non-zero values of the sea-ice concentration - in the construction of the $Q_{rp}$ fields in the nudging experiment.

The nudging experiment is also used in the training of the ANN, thus resulting in negligible corrections therein. Additionally, no sea-ice predictors are used. This is because sea surface temperature data beneath sea ice are extrapolated from sea ice concentration data and are less reliable (Rayner et al., 2003).

After a preliminary comparison of different model architectures (not shown), the best-scoring neural network model includes 3 hidden layers (5 total), 256 neurons (considering an input size of 24 features and an output size of 1), and uses the rectified

linear unit (ReLU) activation function in all layers but the last one. All input and output variables were normalized by their global mean and standard deviation. During the training, we used daily means, subsampled every 5 days during the period 2003-2017; while, at the same temporal frequency, the years 2001, 2002, 2017, and 2018 were used for validation within the ANN training, and 2019-2020 as independent test datasets.

We tested the impact of the correction frequency and training dataset's timescale in preliminary experiments with the NEMO model and the online ANN's correction of $Q_{rp}$; we aimed to assess the impact of high temporal frequency in the inference step, ranging from monthly to daily sets of predictors-corrections, and investigate the impact of the frequency of the inference step in NEMO from daily to 3-hourly. The results are summarized in Figure 1, in terms of global sea surface temperature RMSE, during the independent verification period 2019-2020. We progressively improve the performance of the ANN-based inference in NEMO, closely approaching the REF experiment with SST nudging, by increasing both the temporal frequency of the predictor-correction datasets and the temporal frequency of the inference step. The best results are obtained for daily sets of predictors-corrections and 3-hourly inference step frequency. Note that we cannot increase it further, because 3 hours is the frequency of the surface boundary condition calculation in our configuration of NEMO.

Next, we show in Figure 2 the error maps of the inferred heat flux correction from test (i.e., independent) data. The Normalized RMSE (panel a) shows errors smaller than 10%, and on average equal to 4% (corresponding to 1.36 W m-2); while errors peak in areas of large mesoscale activity (western boundary currents and the Antarctic Circumpolar Current, ACC), there exist other non-obvious local peaks. The systematic error of the ANN reconstructions is very low (panel c), generally not exceeding 0.7 W m-2, indicating that the RMSE is explained primarily by random errors (panel d shows the standard deviation of the differences). Note that the grid-point-wise correction implies that the smoothness of the ANN-based correction depends on that of the predictors, i.e. the model fields; we verified the high consistency between the original output and the NN-inferred one even in individual snapshots (not shown).

Table 1 reports the list of predictors, grouped into categories, together with their impact in terms of Variable Importance Scores (VIS). VIS for the predictors is calculated through the permutation-based method of Fisher et al. (2019), through the *vip* R package (Greenwell and Boehmke, 2020), applied on the entire pre-trained model, or pointwise for each model grid-point (see Table 1's caption for details). The Total VIS refers to the VIS over the full columnar ANN model, while the local VIS is calculated for each gridpoint by fixing the longitude-latitude pair to the corresponding gridpoint. The different VIS results respond to different questions, i.e. the Total VIS indicates the global impact of each predictor on the final ANN. Diagnosing local VIS allows investigating the regional patterns of variable impact, and Table 1 also reports its spatial average values.

The explainability results for the entire pre-trained model suggest a large impact from static data, wind forcing, and temperature; a significant impact from the heat flux components, and a relatively smaller impact from salinity, freshwater fluxes, and ocean stratification. There may exist, however, non-exclusive attributions of the errors to the predictors, as important correlations between parameters exist. For instance, VIS for temperature may partly indicate errors in climatological flux (due to the climatological state of the sea surface) or air-sea heat flux (e.g., the upward longwave heat flux); the wind forcing may also explain systematic errors in, e.g., latent heat flux; and so on for other correlated fields. Due to the strong

multivariate nonlinearities of air-sea interactions (e.g., wind stress depending on both near-surface winds and local temperatures via nonlinear bulk formulas), these correlations are not reducible, and we take the practical approach to diagnose their impact as it comes from the VIS metrics. In some cases predictors respond to very similar processes, although not identical either (e.g., wind stress and wind speed differ from the use of sea surface currents in the former, etc.).

Figure 3 shows the most impactful predictors as a function of longitude and latitude (both individual predictors and categories). This indicates that in most of the global ocean, the most important predictor is associated with wind forcing (either wind speed or stress). Interestingly, mesoscale active areas (e.g., western boundary current regions and the Antarctic Circumpolar Current) exhibit turbulent heat fluxes (latent and sensible heat) as the dominant predictor, consistently with the large influence of ocean mesoscale dynamics in air-sea exchanges therein (see, e.g., Frolov et al., 2021). In many coastal regions, the most important predictor is associated with freshwater fluxes. Only a few grid points exhibit another dominant predictor.

Figure 4 shows the individual impact of each predictor (in %), disclosing interesting spatial patterns, closely related to physical and dynamical processes. For instance, the mixed layer depth appears important near the Equator, likely related to ENSO variability; precipitation's impact is relevant in correspondence to the ITCZ (Inter-tropical convergence zone) likely due to its possible misplacement, and around the maritime continent. Eastern boundary upwelling systems are impacted by the solar heat flux, and the diurnal and seasonal variability (namely, the SST diurnal amplitude and the day of the year, respectively). The salinity flux is relevant over marginal ice zones, in both polar regions, associated with ice-ocean freshwater and heat exchanges; river runoff impacts the flux errors in the proximity of the shorelines.

## 2.3 Experimental setup

Several experiments were run with the NEMO ocean model equipped with new functionalities, to store in a rolling array the predictors at the desired temporal frequency (see section 2.2 and Figure 1). We use an in-house Fortran90 library (see the Code availability section) for online inference from the pre-trained model, given that the NEMO model is coded in Fortran, and this eases the online inference step. In detail, the prediction step is natively implemented in Fortran90 as an additional NEMO module to avoid the need for external software interfaces. The pre-trained model is read in at the beginning of the NEMO model integration; then, every 3 hours, the inference routine is called with the predictors average over the latest 24 hours as input. The inferred corrective flux is then added to the uncorrected (bulk formula-derived) non-solar heat flux component, every 3 hours.

The experiments with the ANN-based heat flux correction, presented hereafter, are named NNC (Neural Network-based Correction) and cover four different scenarios: i) validation in the training phase (self-consistency), i.e. during the period 2002-2018; ii) validation in the test phase (independent verification), i.e. during the period 2019-2020, after the training period; iii) validation in earlier periods, where no dense SST data were available (1961-1979), aiming to test the impact of the new method for retrospective simulations and reanalyses, without any memory in the ocean state initialization; iv) validation in prediction experiments, namely 7-day forecasts initialized every 10 days in 2021 and 2022 from the data assimilation-enabled CIGAR reanalysis (Storto and Yang, 2024) and forced at the sea surface by the ECMWF operational forecasts replacing the ERA5

reanalyses used in the scenarios i), ii) and iii). These setups allow us to provide a full assessment of the methodology for different applications (long- or short-term simulations, historical reanalysis, and operational oceanography).

Further to NNC, we show results from REF (standard SST nudging enabled), CTRL (no corrections), and CLIMC (climatological corrections). The latter corrects the air-sea heat fluxes with a monthly climatology of corrections derived from the REF experiment, representing a linear benchmark for the methodology used in the NNC experiment.

## 3 Results

### 3.1 Contemporary simulations

The reconstruction of corrective fluxes with the pre-trained model is shown in Figure 5, which indicates the close correspondence between the SST nudging-derived and the neural network inferred fields, during the full period 2001-2020. Large corrections occur in mesoscale active areas (with large but not exclusive role of turbulent heat fluxes, see Figures 3 and 4), the North Atlantic subpolar gyre (with significant role of freshwater-related predictors, see Figure 4), in the Tropical and Southern Oceans. Signs are in general reversed in the Northern and Southern Hemispheres during the winter and summer seasons (namely, the non-solar heat fluxes are underestimated in wintertime and over-estimated in summertime, because of generally cold and warm biases of sea surface temperature, respectively). The seasonality of the corrections in deep convection areas suggests also systematic misrepresentation of convective processes therein, with much too deep mixed layer in the North Atlantic oceans, and more complex patterns in the Southern Ocean and ACC region.

The application of the correction leads to satisfying bias correction during the independent verification period 2019-2020, as shown in Figure 6. Large negative biases in the Gulf Stream, Kuroshio Extension, and central Tropical Pacific, plus locally in the Southern Ocean, present in CTRL are equally mitigated in REF and NNC, and likewise for warm biases in the eastern regions of Tropical basins, in the Indian Ocean, and locally elsewhere. Over the mid-latitudes, SST biases approach zero, while elsewhere the remaining biases that the SST data ingestion was not able to mitigate in the REF experiment are reproduced also in the NNC experiment. The global mean absolute error (MAE) over 2019-2020 decreases from 0.37°C in CTRL to 0.20°C and 0.19°C in NNC and REF, respectively, while CLIMC exhibits a MAE of 0.23°C. Differences between NNC and REF experiments are very small and limited only to polar areas (north of 60°N and south of 60°S), where the NNC corrections are small by construction.

The effects of the correction are also well reproduced in the ocean stratification, shown in Figure 7 in terms of mixed layer depth differences in March and September 2020 compared to the CTRL experiment. Either the SST assimilation or the neural network-based heat flux corrections induce an identical shift in the deep convection areas; in the Southern Ocean, during September 2020, a westward shift is visible in the Pacific sector; other local adjustments are visible in both the Atlantic and Indian sectors of the ACC region. Adjustments are also visible in the Atlantic subpolar gyre, where enhanced convection appears in the Iceland basin and Irminger Sea, equally present in both REF and NNC experiments, along with attenuated mixing south of the Labrador Sea.

The global ocean heat content (OHC) anomaly interannual variations are visible in Figure 8 and show that NNC and REF lead to the same linear trends, and seasonal and interannual variations. Neglecting air-sea heat flux corrections in CTRL produces underestimated global ocean warming (0.15 W m$^{-2}$), which is identically corrected in NNC and REF (0.41 and 0.43 W m$^{-2}$, respectively). Using climatological corrections only partly mitigates the warming under-estimation (0.33 W m$^{-2}$), resulting in an intermediate solution. The correlation of OHC anomalies with respect to independent datasets such as the CIGAR reanalysis is also equally improved (from 0.48 in CTRL to 0.92 in NNC and REF). This suggests that the subsurface signature of the correction method is identical to the original nudging experiment.

Similarly, the global overturning circulation (Figure 9) shows again the same behavior for the REF and NNC experiments, indicating that also the dynamical signature of our approach provides the same results as in the assimilation experiment REF. The assimilation of the SST observations in REF reduces the North-South Hemisphere contrast of the overturning circulation (panel b in Figure 9), which is equally found in NNC.

Finally, the impact is evaluated against fully independent data, namely in-situ profiles extracted from the UKMO EN4 dataset (Good et al., 2013), during the period 2019-2020. This is shown in Figure 10 (left panels) where the RMSE of CTRL is shown, together with the differences of RMSE between REF or NNC minus CTRL. Negative (positive) values indicate an improvement (deterioration) borne by the correction method. The figure indicates the comparable impact of the SST nudging and the neural network correction on reducing the errors in the subtropical and mid-latitude regions, with the sub-surface Tropics less impacted by the corrections.

## 3.2 Retrospective simulations

Retrospective simulations were conducted to evaluate the potential of the method for long-term historical simulations, e.g., for OMIP- and CMIP-like exercises, and in multi-decadal reanalyses where the paucity of observation data in early periods limits the impact of conventional data assimilation and cannot take advantage of space-borne satellite measurements of SST. To this end, the same set of experiments presented earlier is performed for the period 1961-1979, initialized by the same initial conditions in 1961 taken from previous simulations.

We show the impact of NNC in terms of RMSE decrease versus the CTRL experiment in Figure 10, compared also, as an independent reference, to the CIGAR reanalysis (Storto and Yang, 2024) that assimilates all in-situ surface and sub-surface observations, and includes a deep-ocean large-scale bias-correction scheme. Improvements are present everywhere in NNC, except in the high-latitude 100-300 m depth layer, although smaller than CIGAR, especially in the Northern Hemisphere. The total average improvement (RMSE decrease) in the top 300 m of depth, compared to CTRL, is 22% for CIGAR and 7% for NNC, meaning that about one-third of the improvement borne by assimilating the full oceanic observing network and applying conventional bias-correction is achieved with the neural network-based correction. The improvement is remarkable at all latitudes, also in the sub-surface Tropical region where the correction over the more recent years 2019-2020 failed to provide significant improvement (middle-left panel in Figure 10). Finally, Figure 10 also shows a 1961-1979 experiment with nudging

to COBE SST (Ishii et al., 2005; experiment NDG); the results show a positive impact of the nudging scheme, although it is generally smaller than the use of ANN to correct the sea surface biases.

## 3.3 Forecast experiments

Forecast experiments are set up with the same model configuration but different initialization and forcing as detailed in Section 2.3. The correction is then applied online within the forecasts, as a proof-of-concept for operational purposes. Unlike the nudging scheme that depends on observational data and cannot be used in forecasts, the ANN-based correction depends only on the oceanic and atmospheric states, thus it can be adopted in operational forecasting systems.

Sea surface temperature errors (verified against mapped satellite data from DOISST v2.1, Huang et al., 2021) as a function of forecast lead time (Figure 11) indicate that NNC provides improvements comparable to nudging – shown as a benchmark – except in the Northern Hemisphere Extratropics, likely because of the intense mesoscale variability. The climatological corrections (CLIMC) fail to improve the CTRL experiment, as they cannot adapt to the variations of the atmospheric forcing in the forecast experiments. Compared to CTRL and considering the errors given by the climatology (dashed lines in the panels of Figure 11), the NNC scheme extends the horizon of useful forecasts by about 1 day in all regions. The impact of the method increases with the forecast lead time, suggesting that the approach might be fruitfully applied in long-range forecasting systems (sub-seasonal and beyond), although it should be demonstrated that coupled feedbacks in the case of Earth System models do not compromise the algorithm.

Similar results are found in the verification against in-situ profiles for the upper ocean (sea surface to 50 m of depth), shown in Figure 12. The top 50 m exhibit significant improvement in the Southern Extratropics and the Tropics, with an improvement borne by NNC increasing with forecast lead time. In the Northern Extratropics, the ANN correction leads to negligible improvements.

## 4 Summary and Discussion

In this work, we propose an algorithm to correct air-sea heat fluxes by letting a neural network pre-trained model learn the relationships between ocean and atmospheric state predictors and heat flux corrective terms, estimated from a previous experiment that adopted sea surface temperature nudging to estimate and apply such terms. The predictors include several oceanic and atmospheric variables representative of the heat, freshwater, momentum fluxes, ocean temperature and salinity, and stratification. A feed-forward column neural network architecture is adopted, and the NEMO ocean general circulation model is augmented with online inference capability to allow collecting predictors and inferring corrections of the air-sea heat fluxes, based on the pre-trained model. Variable Importance Scores indicate the large impact that wind forcing has on the errors in most parts of the global ocean, with other variables dominating locally, e.g. turbulent fluxes in mesoscale active areas and freshwater fluxes near the coasts.

The online use of the correction in the experiments indicates that the approach successfully reproduces the surface, sub-surface, and dynamical signature of the SST correction, even beyond the training data period. The corrections are by construction representative of all SST errors that are corrected in the nudging experiments, namely not only the heat flux inaccuracies but also other errors related for instance to vertical mixing, solar radiation penetration, etc.

Next, the approach is demonstrated in early periods (1960s and 1970s) where surface temperature data are sparse, to mimic a long-term simulation or reanalysis application. In this context, the methodology provides a significant improvement in subsurface temperature errors, roughly equal to one-third of the improvement in a corresponding reanalysis system where all available observations are directly assimilated.

We also showcase the method in short-range prediction experiments, where observations cannot be used to correct the forecast
step; the methodology is proven to significantly reduce surface and subsurface temperature errors, at a negligible extra computational cost and without the use of any observational information, increasing the SST predictability of about 1 day at all latitudes. Subsurface errors are also mitigated everywhere except in the Northern Extratropics.

We have demonstrated as well the significant impact of online inference, which allows high-frequency (3-hourly) updates of the correcting fields. For this reason, testing different model architectures, e.g., those relying on convolutional layers, which
require MPI communication across NEMO domains inside the convolutional filters, was technically complex and demanding. It is not obvious whether convolutional layers are beneficial compared to grid-point-wise corrections (see, e.g., different conclusions in Chen et al., 2022; Chapman and Berner, 2024), as the potential advantage of retaining horizontal patterns is balanced by the computational needs of coarsening the spatial resolution. In the future, more sophisticated inference libraries and tools for online prediction are expected to be available, paving the way for testing different neural network architectures.

While ANNs cannot provide improvements compared to the data assimilative experiments that are learning from, their use is appealing for several different applications that cannot rely on the observational input: simulations and projections, and multi-decadal reanalyses spanning early periods with scarce observations, as demonstrated in this article. However, applying this method within a coupled ocean-atmosphere model may benefit climate drift correction (e.g., Gupta et al., 2013) but introduces additional complexities due to nonlinear coupled feedback. In a coupled model, the atmosphere could respond to modified
fluxes in a nonlinear and potentially unpredictable manner. Heat flux corrections that work well in an uncoupled system may introduce unintended biases when the atmosphere reacts dynamically, potentially leading to unrealistic SST adjustments. Atmospheric variability (e.g., cloud cover, wind stress, and humidity) will alter in response to changes in SST, which could impact the efficacy of the ANN-based correction. Corrections applied at short timescales may also have long-term impacts on coupled modes of variability (e.g., ENSO, MJO).

To make the ANN approach more suitable for coupled applications, it could be retrained using data from coupled model reanalyses (e.g., CMIP simulations or CERA reanalysis datasets, Laloyaux et. al., 2016), or observations (e.g., Zhou et al., 2024). This would allow the ANN to learn heat flux corrections in a system where atmospheric responses are accounted for, in analogy with flux correction or flux adjustment techniques (e.g., Sausen et al., 1988). The ANN-based correction could be implemented to maintain the overall coupled energy balance while addressing systematic errors.

Additionally, the algorithm could include corrections also to freshwater and momentum fluxes, subject to long and reliable datasets of e.g. sea surface salinity and currents to first estimate their corrections, whose availability is limited now.

The method represents the first attempt to leverage data assimilation correction increments, in this case from SST nudging, to learn systematic errors in ocean models. It is also expected that higher-resolution implementations than that presented here may further benefit from the ANN compared to climatological corrections, due to their higher spatial and temporal variability.

While providing good results in hindcast mode compared to the control experiment, climatological corrections fail in predictive experiments without proper retuning and re-computation through computationally expensive re-forecast experiments. This in turn suggests the possibility of extending the approach for calibrating forecasts without the need for long re-forecasts.

Further extension of the approach will consider full column increments for three-dimensional corrections, not only associated with heat fluxes but also vertical physics and model parametrizations; while this has been proven successful in atmospheric

(e.g., Chen et al., 2022) and sea-ice (Gregory et al., 2023) applications, ocean implementations are more challenging due to scarce observing networks in the ocean interior, potentially hampering the use of analysis increments at depth, which is an active area of investigation at the moment.

**Code availability**

The NEMO model is available through the official website https://www.nemo-ocean.eu; version 4.0.7, used in this study, can be downloaded at http://forge.ipsl.jussieu.fr/nemo/changeset/15814/NEMO/releases/r4.0/r4.0-HEAD?old_path=%2F&format=zip. In-house modifications to the NEMO model code as used in the experiments presented here – including the module for ANN-based corrections, plus other modifications – are available as a git repository at

https://baltig.cnr.it/nemo_ismar-rm/nemo_4.0.7/-/tree/3.0?ref_type=tags There are several additional modifications than just the ANN-correction routine, which can be found in the *dnnqcorr* module. The ANN correction routine can be however isolated taking only the modules dnnqcorr.F90, and adding the call to dnn_qcorr in sbcmod.

The library for online inference in Fortran90 used in our NEMO experiments is available as a git repository at

https://baltig.cnr.it/andrea.storto/nnt4nemo/-/tree/main/F90_Inference It includes the ANNIF module for reading pre-trained ANN in NetCDF format, plus inference routines and their tangent-linear and adjoint versions.

The frozen version of both source codes, together with the scripts and the data to analyze and plot the results presented in the figures, are available as a dataset at Zenodo as https://doi.org/10.5281/zenodo.13380698.

**Data availability**

Atmospheric fields from ECMWF to force the ocean have been taken from the Climate Data Store (CDS, https://cds.climate.copernicus.eu) archive (ERA5) and the operational archive (operational forecasts, see www.ecmwf.int). SST data are available from the U.K. Met Office Hadley Centre (https://www.metoffice.gov.uk/hadobs/hadisst). For verification purposes, we used SST analyses from the NOAA DOISSTv2 dataset

(https://psl.noaa.gov/data/gridded/data.noaa.oisst.v2.highres.html) and in-situ profiles from the U.K. Met Office EN4 dataset (https://www.metoffice.gov.uk/hadobs/en4/download-en4-2-2.html).

**Author contribution**

AS and SF have designed the methodology; AS has coded the methodology and run the experiments; LS and CY have provided guidance on the model developments and assessment results. AS drafted the initial version of the manuscript; all coauthors

have discussed the results and revised the paper.

**Competing interests**

The authors declare that they have no conflict of interest.

**Acknowledgments**

**Financial Support**

This research is supported by the Research Program CN00000013 "National Centre for HPC, Big Data and Quantum Computing" Directorial Decree (grant no. 1031 of 17 June 2022) from the resources of the PNRR MUR – M4C2 – Investment 1.4 – "National Centers" Directorial Decree (grant no. 3138 of 16 December 2021). NOAA PSL provided financial support for Dr. Storto's visit to the lab in Summer of 2023.

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

| Category | Predictors | Total VIS (%) | Gridpoint-averaged Local VIS (%) |
|---|---|---|---|
| Stationary | Lon, Lat, Time | 30 | 7 |
| Temperature | SST, OHC, SST_da | 22 | 12 |
| Salinity | SSS, OSC | 4 | 1 |
| Heat flux | Qlat, Qsen, Qlw, Qsw, Qemp | 11 | 18 |
| Freshwater flux | Precip, Runoff, Salt flux, | 6 | 16 |
| Wind forcing | Stress modulo, Wind speed, SSH | 26 | 44 |
| MLD | MLD, MLD_da | 1 | 2 |


**Table 1. List of predictors, grouped by categories, with their aggregated Variable Importance Score (VIS), given as percent impact, both as the impact on the pre-trained model (Total VIS), and averaged over the global domain from the pointwise application (Gridpoint-averaged Local VIS). MLD: mixed layer depth; OHC: ocean heat content; SSS: sea surface salinity; OSC: ocean salt content; SSH: sea surface height; the suffix _da refers to diurnal amplitude.**


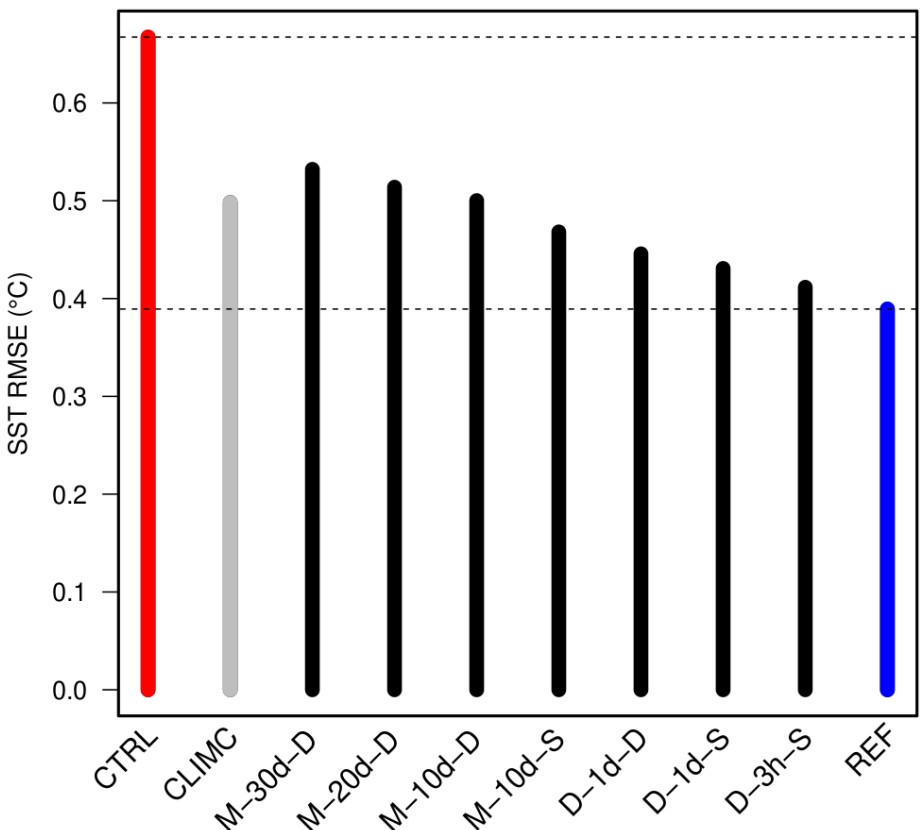

**Figure 1. Sea surface temperature globally averaged RMSE for preliminary experiments, over the independent verification period 2019-2020. M-\* experiments and D-\* experiments refer to the use of monthly versus daily averaged nudging increments in the ANN training; the second string in the experiment name (30d, 20d, …, 3h) refers to the length of the predictor rolling archive; the last letter refers to the frequency of the update in the online experiments ("D" as daily, "S" as sub-daily, namely every 3 hours).**


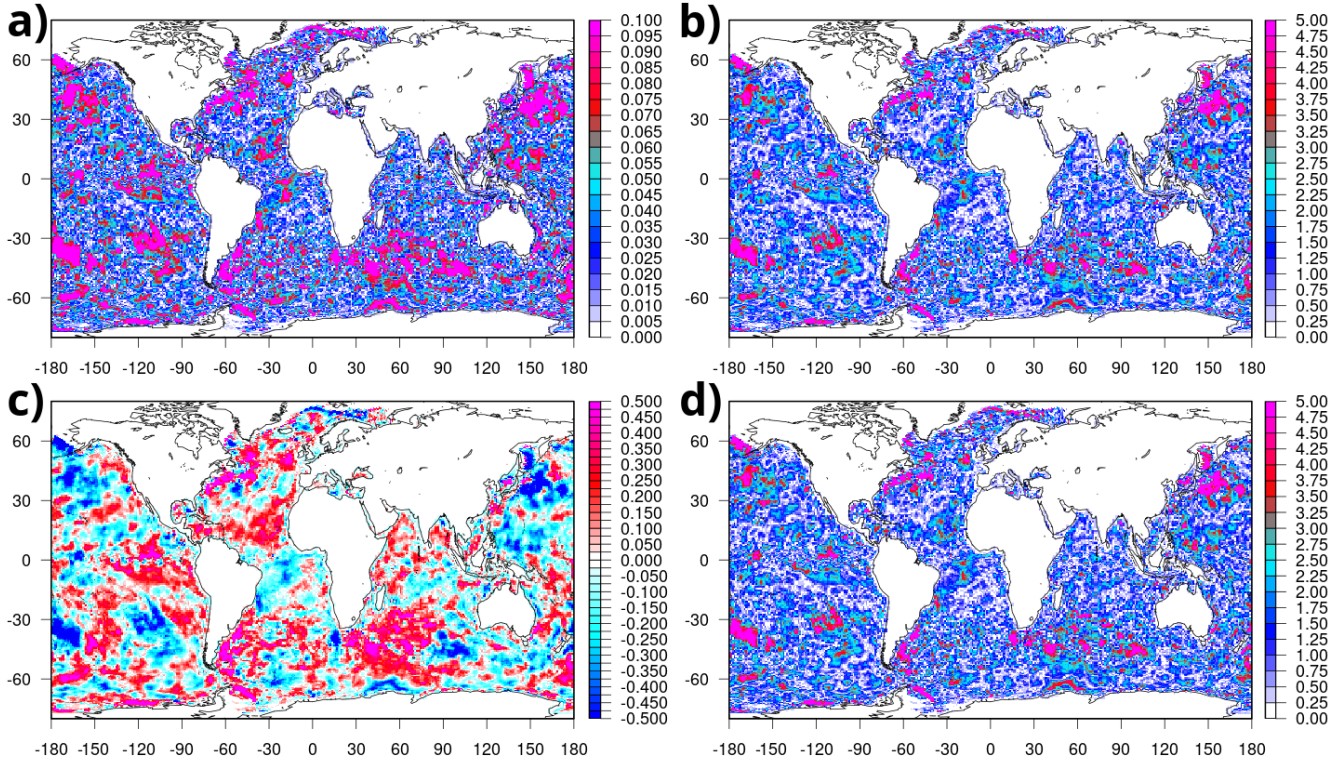

**Figure 2. Error maps of the reconstructed heat flux correction with test data, namely independent data from the training during**
**2019-2020. a) normalized RMSE (dimensionless); b) RMSE in units of heat flux (W m⁻²); c) bias (W m⁻²); d) standard deviation of**
**the differences between original heat flux correction and those reconstructed with the ANN (W m⁻²).**

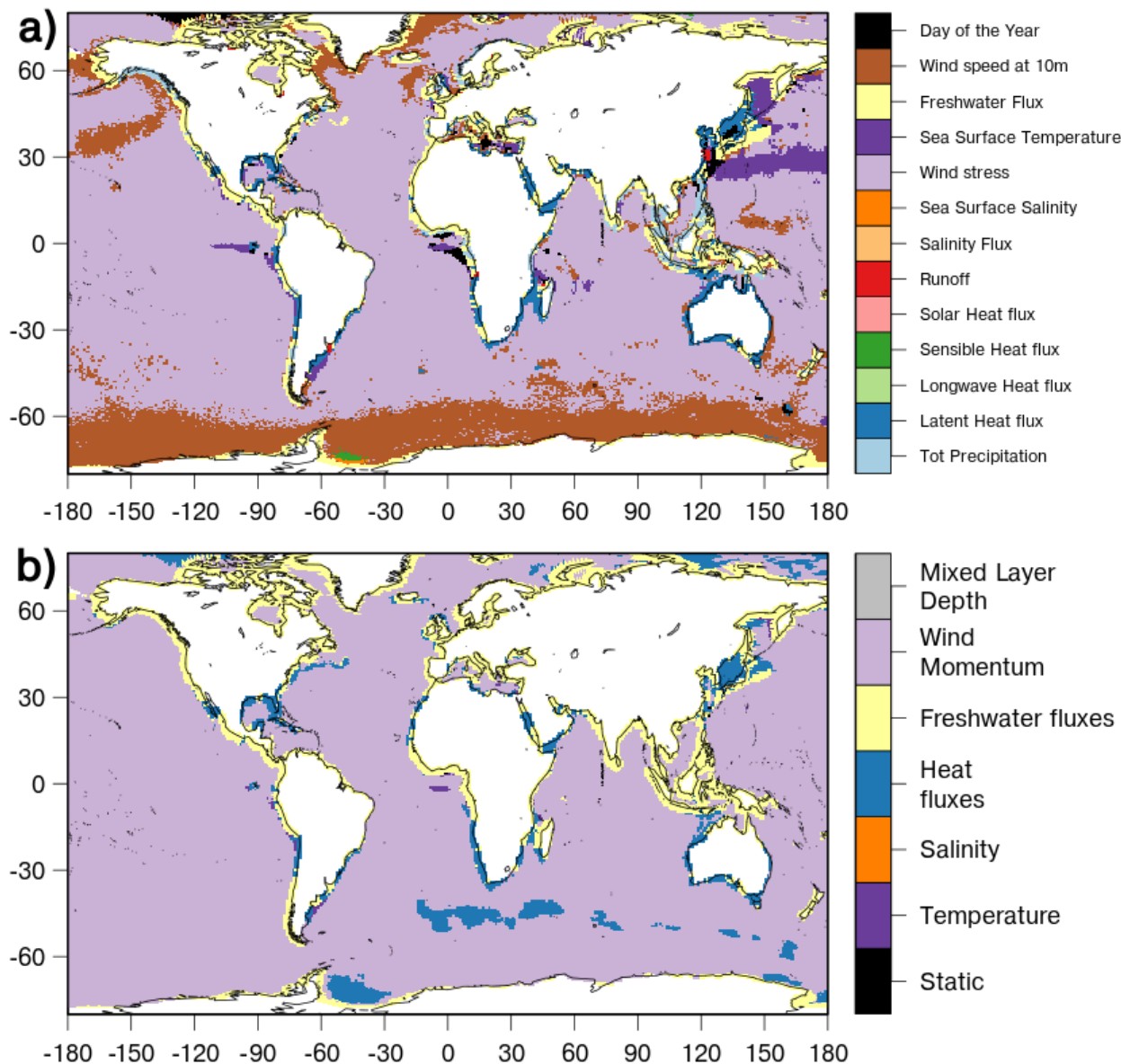


**Figure 3. Dominant predictors identified by Variable Importance Scores (by individual predictor, panel a, and by predictor categories, panel b), from the optimal pre-trained model described in the text. The predictors' list is as in Table 1, but for the sake of clarity only those predictors with at least one dominant gridpoint are considered in panel a).**

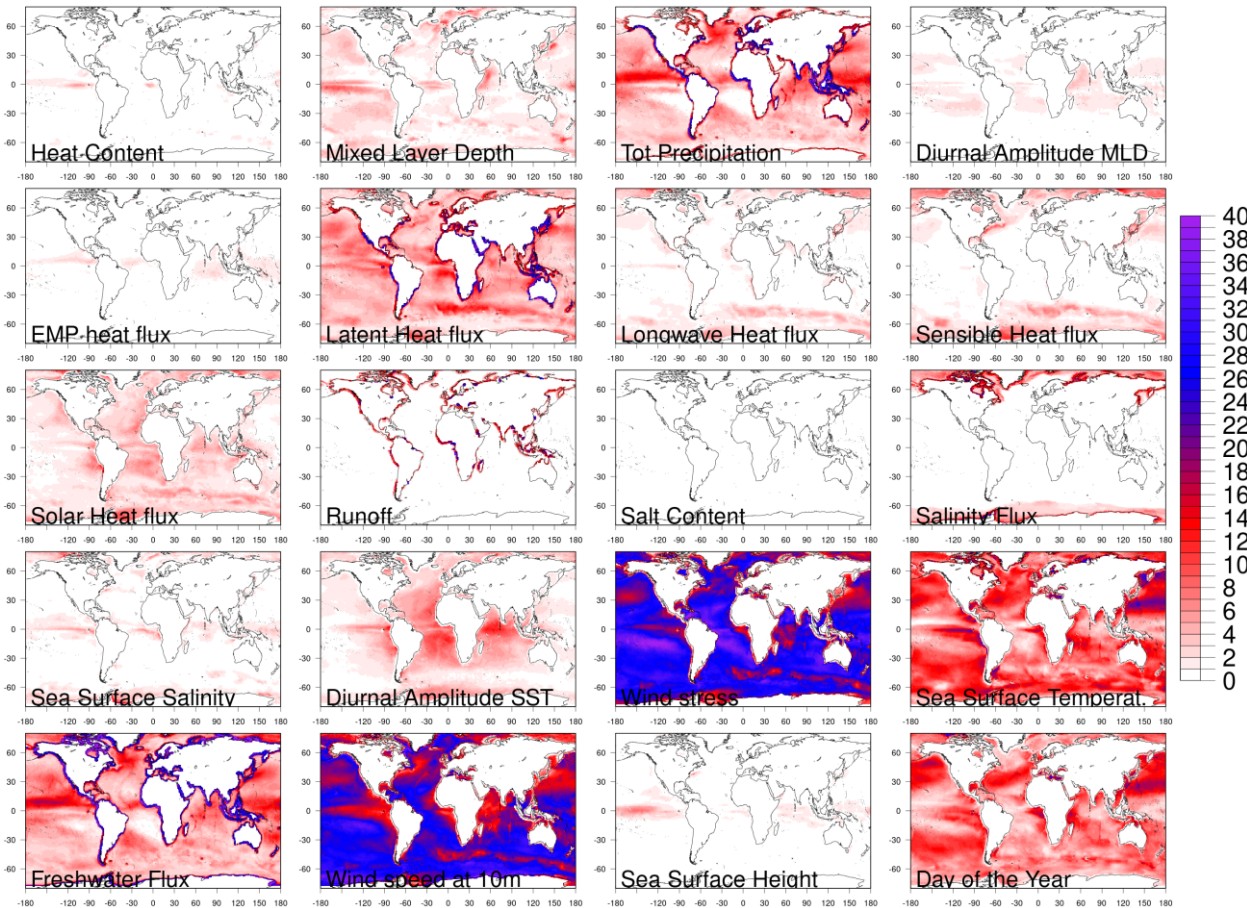

**Figure 4. Variable Importance Score (VIS, in % values) for each predictor used in the neural network pre-trained model, as a function of grid point (namely, fixing the values of longitude and latitude). VIS maps are used to locally attribute different sources of air-sea heat flux errors to the predictors.**

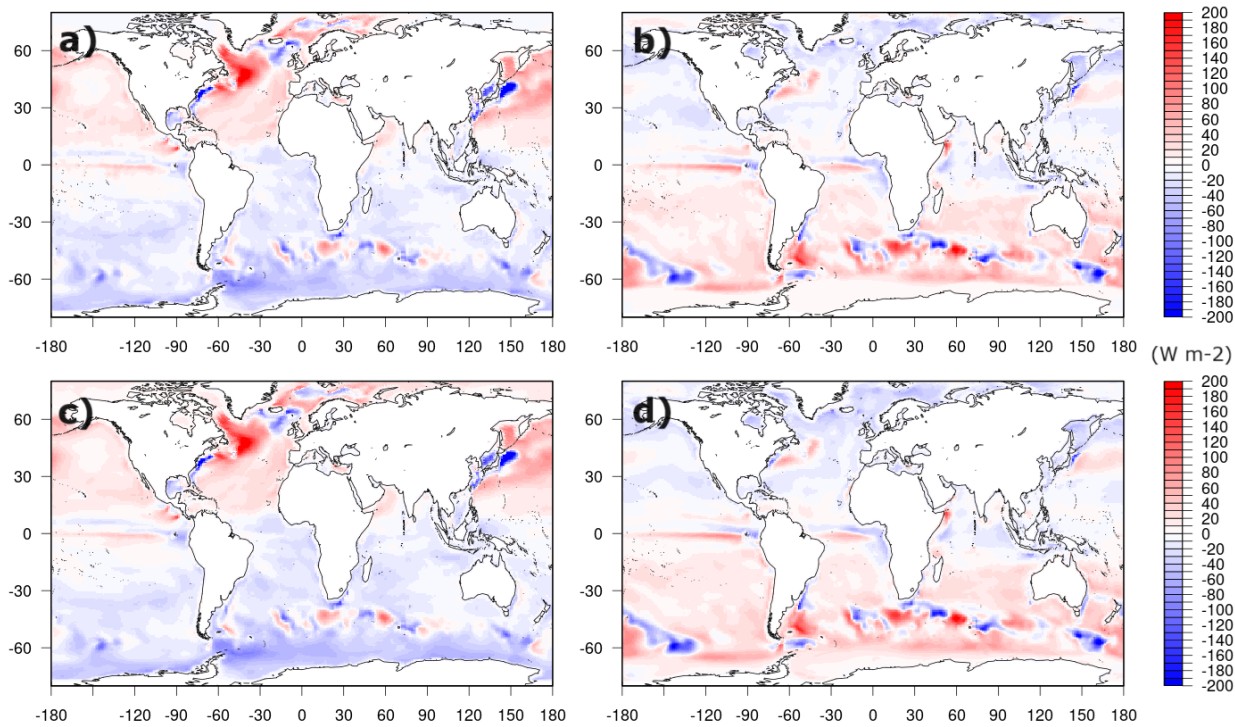

**Figure 5. Reconstructed heat flux correction fields versus the original ones from the REF (a, b) and NNC (c, d) experiments, for JFM (a, c) and JJA (b, d) seasonal climatologies, during the 2002-2020 period.**

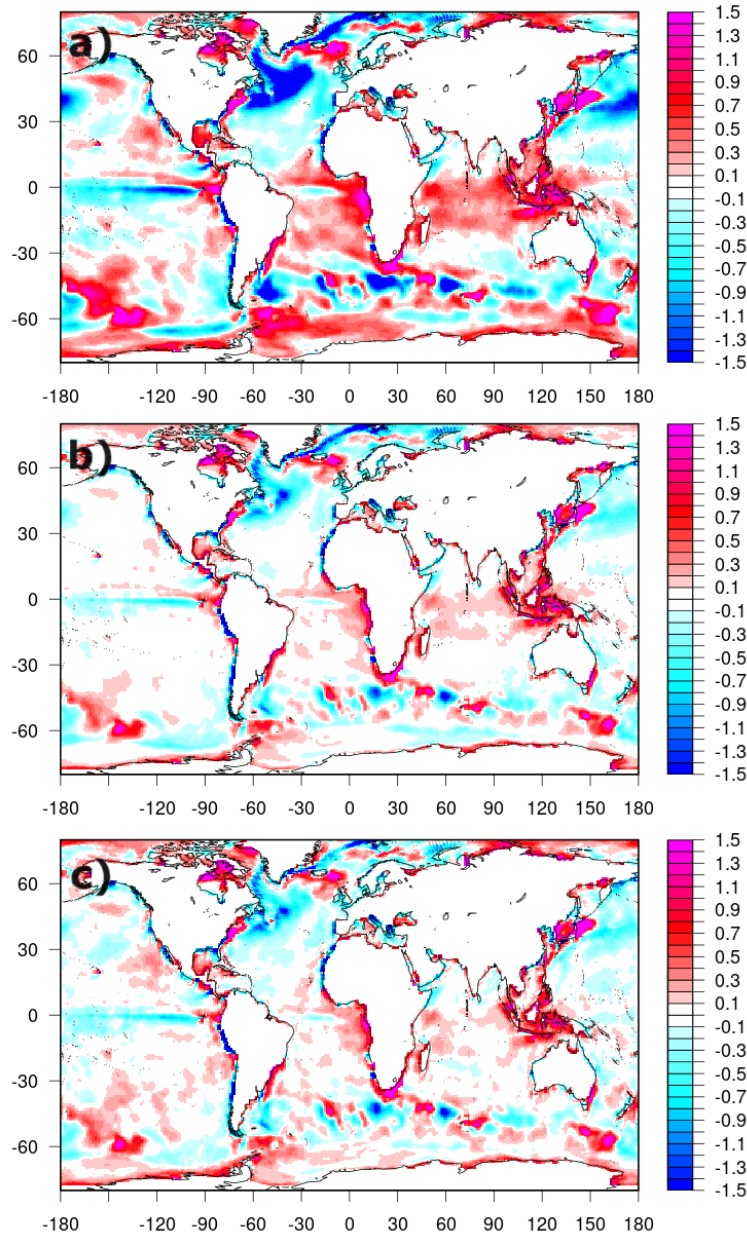


**Figure 6. SST Bias over the independent period 2019-2020 against the SST observations (from UKMO HadISST), for the three experiments CTRL (panel a), REF (panel b), and NNC (panel c).**

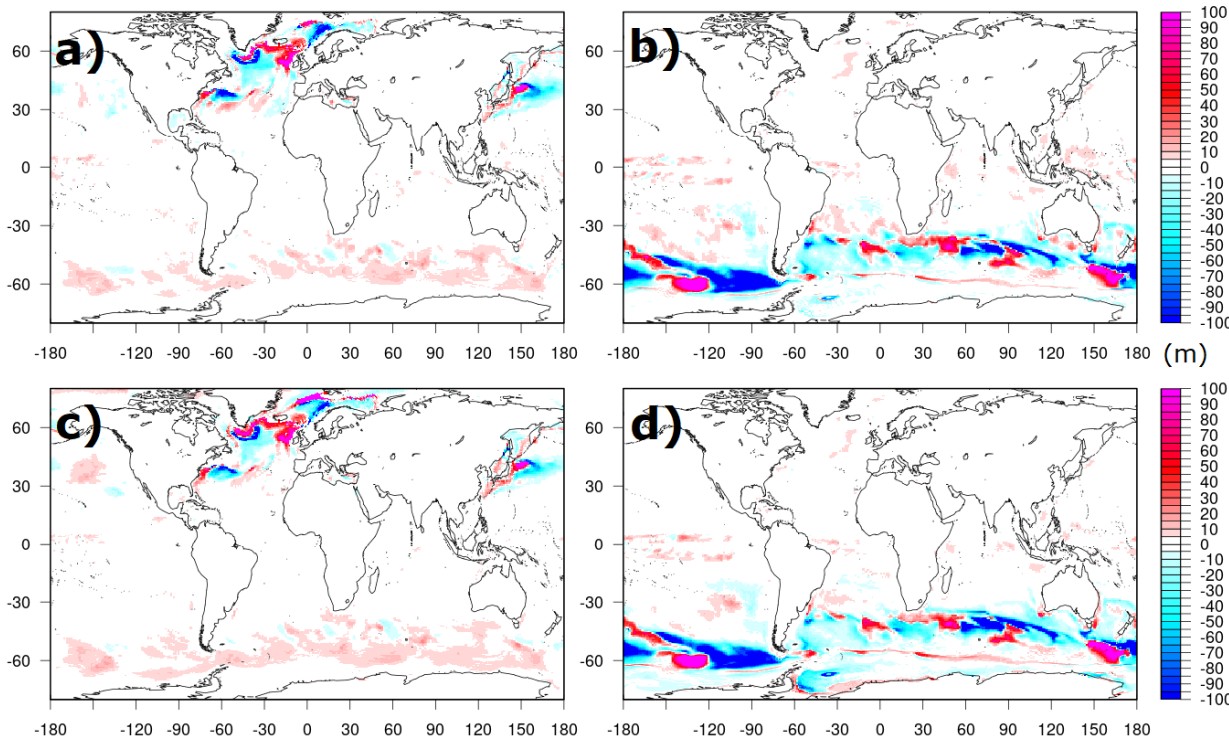


**Figure 7. Mixed layer depth differences with respect to the CTRL experiment during March 2020 (panels a, c) and September 2020 (panels b, d), for experiments REF (panels a, b) and NNC (panels c, d).**

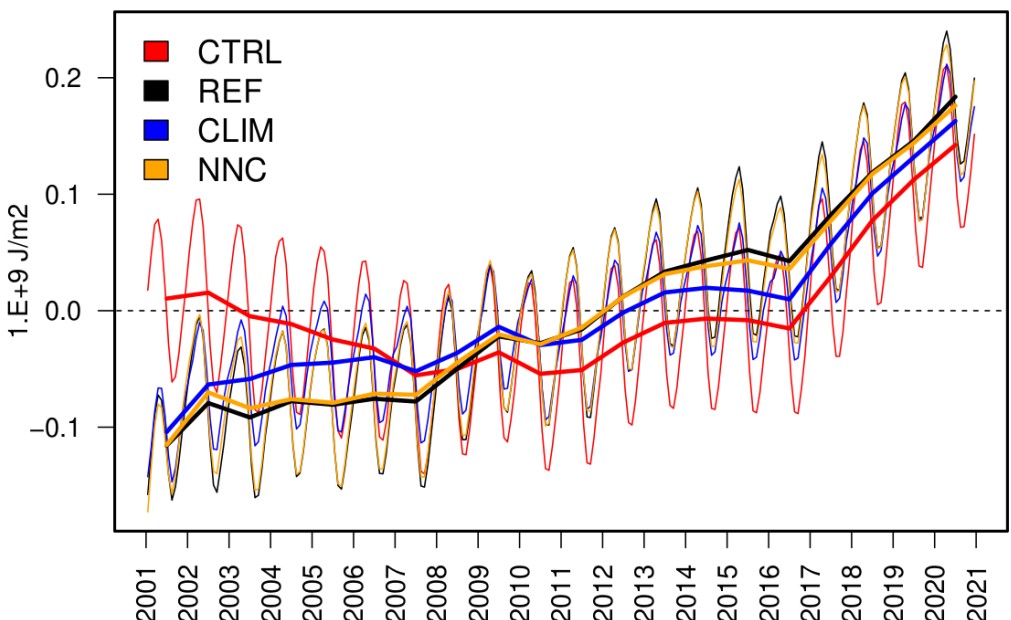

**Figure 8.** Global ocean heat content anomaly vertically integrated over the period 2001-2020 for the four experiments presented in the text, as monthly (thin lines) and yearly (thick lines) means.


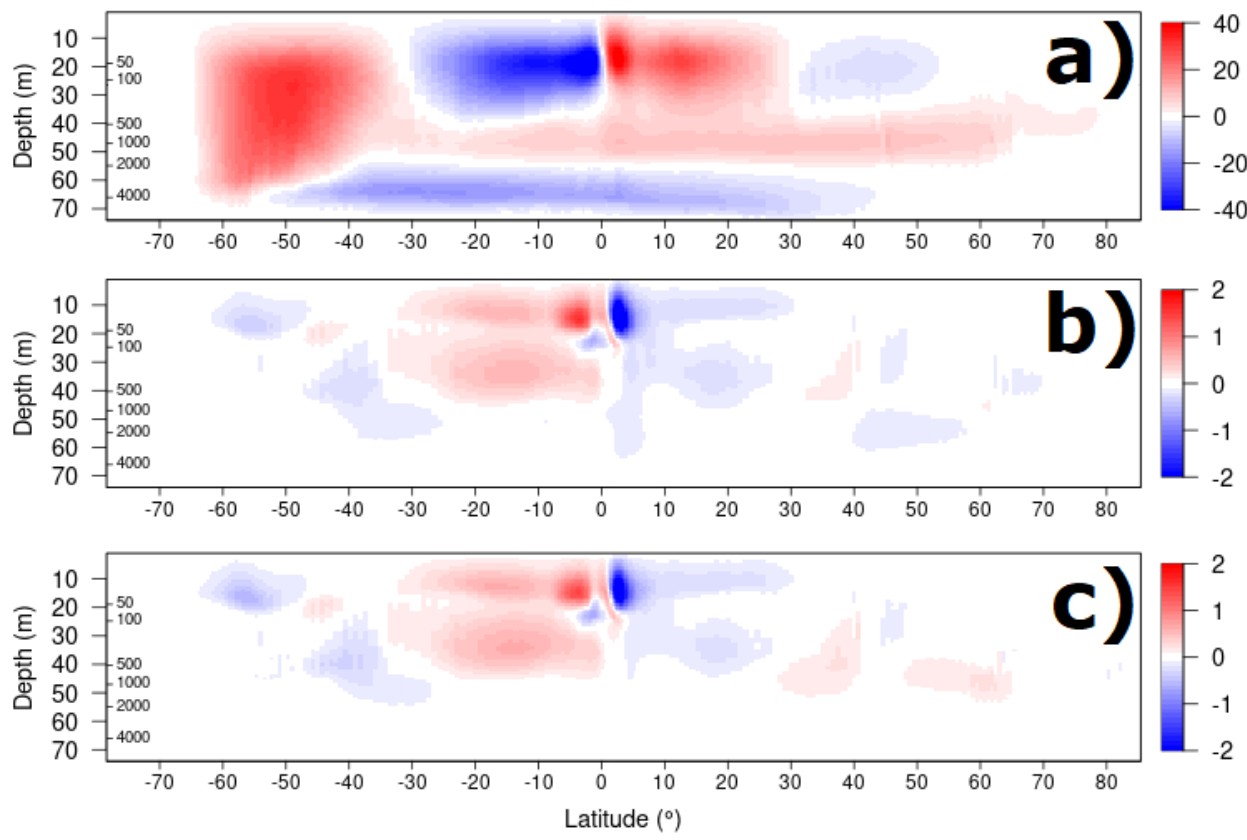

**Figure 9. Reconstructed global overturning circulation (in Sverdrups, with 1 Sv = 1E+6 m3 s-1) for the CTRL (a), and as a difference between CTRL and REF (b), or NNC (c) experiments.**

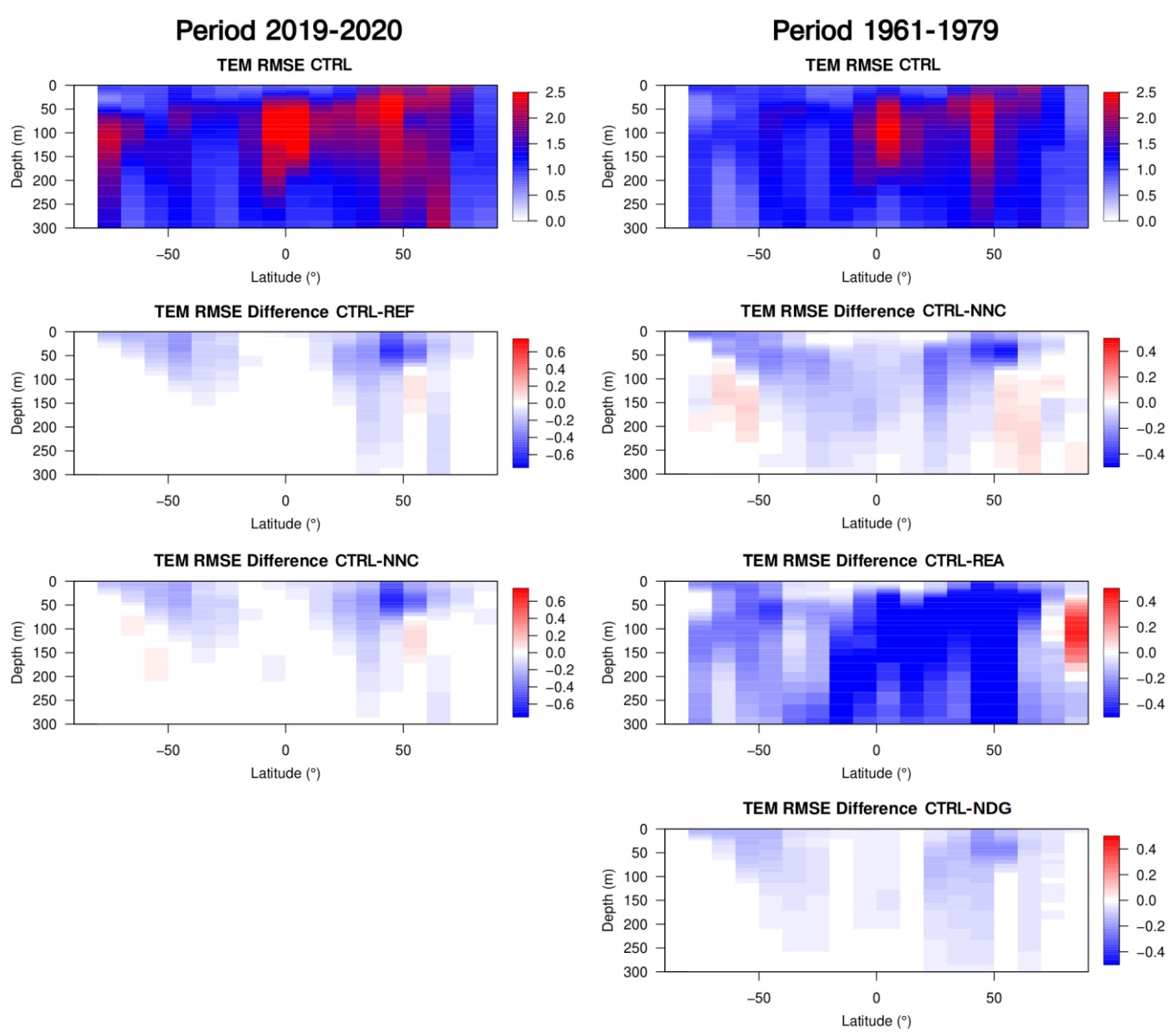

**Figure 10. Temperature RMSE as a function of latitude and depth for the CTRL experiments (top panels) and for the period 2019-2020 (left) and 1961-1979 (right), and differences between CTRL and REF or NNC (left) and NNC or REA or NDG (right) for their respective periods. REA is the CIGAR reanalysis, while NDG only ingests mapped in-situ SST data from the COBE dataset (Ishii et al., 2005).**

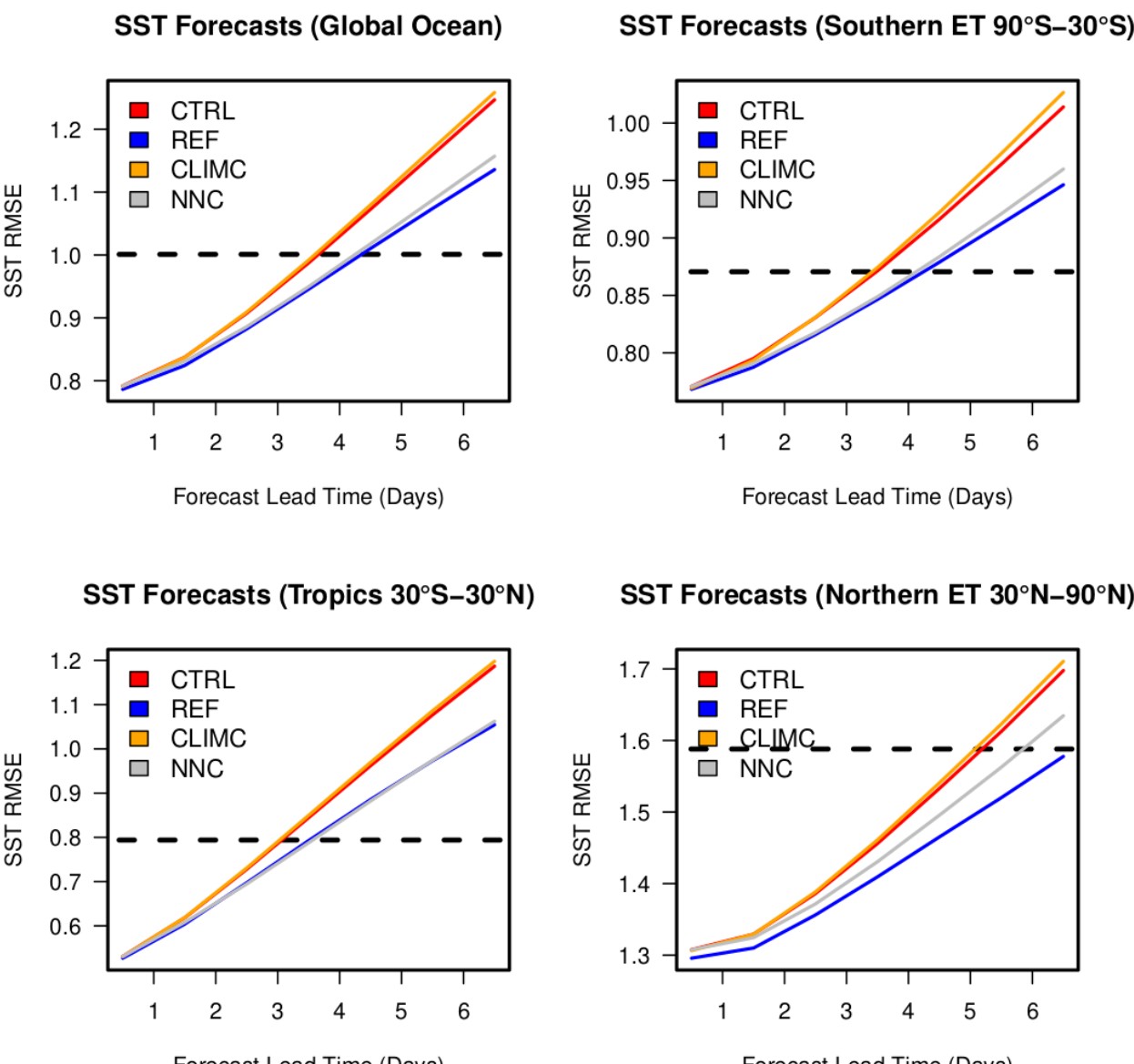

**Figure 11. Forecast skill score metrics (RMSE), for sea surface temperature at different latitudinal bands, as a function of forecast lead time, for the experiments presented in the text. The dashed line corresponds to the RMSE of climatology, i.e. for values of RMSE greater than the climatology the forecasts are not useful. Note that the REF experiment is shown as a benchmark, but its setup cannot be used in operational experiments, as it relies on future observations.**

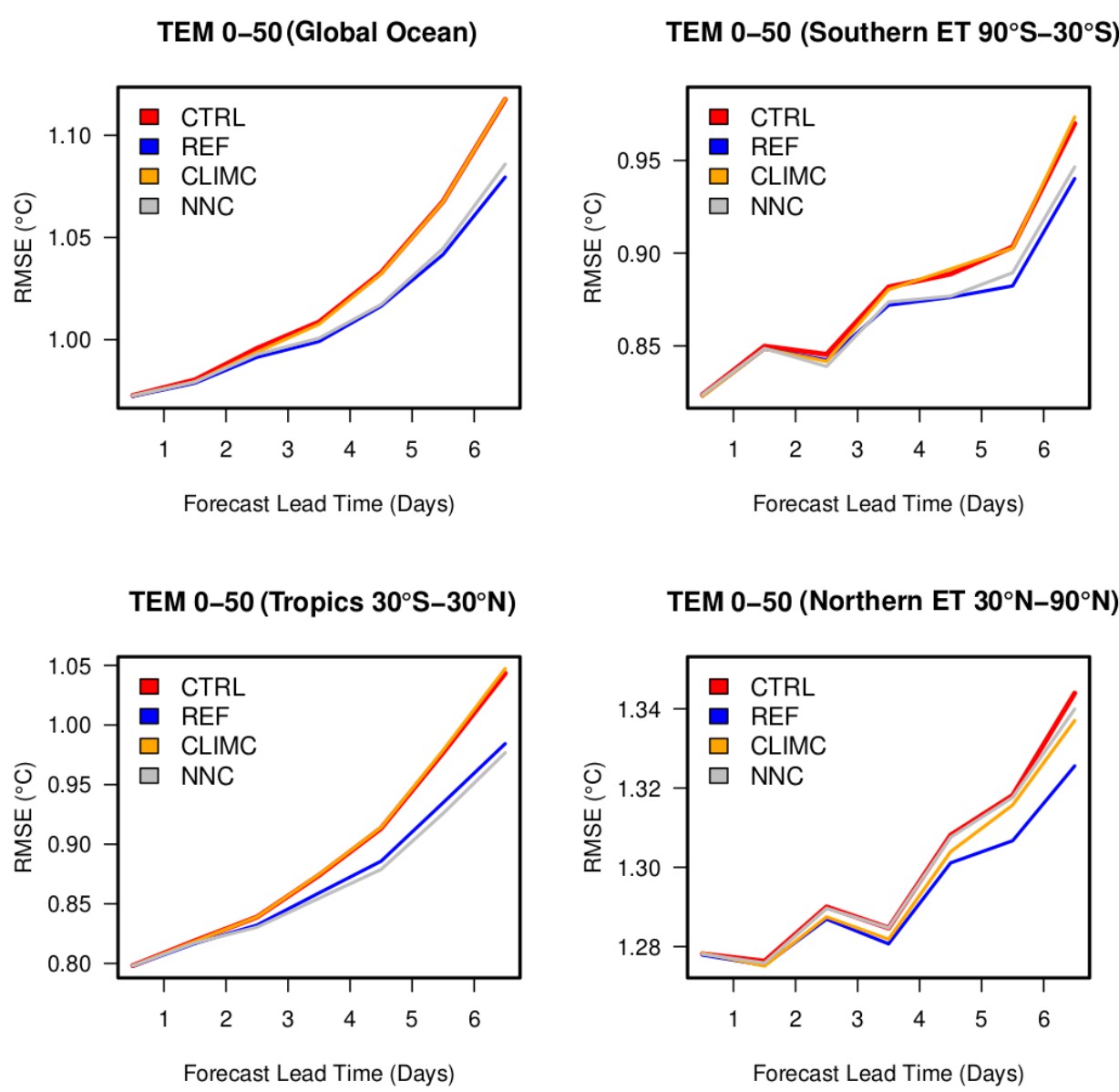

**Figure 12. As Figure 11 but for the verification against in-situ profiles in the top 50 m of depth.**