# Peer review of "Correction of Sea Surface Biases in the NEMO Ocean General Circulation Model Using Neural Networks"

_Geoscientific Model Development, 2024_

## Referee Comment (RC2)

A review of **Correction of Air-Sea Heat Fluxes in the NEMO Ocean General Circulation Model Using Neural Networks**
By Andrea Storto et al.

Submitted to Geoscientific Model Development
Manuscript gmd-2024-185, https://doi.org/10.5194/gmd-2024-185

The authors present a data-driven method for "correcting" the air-sea fluxes in the state-of-the-art NEMO ocean general circulation model. The method is mimicking a sea-surface relaxation scheme that is commonly used in NEMO to constraint the sea-surface temperatures (SSTs) to external datasets, which would most often by remote-sensed satellite products. In practise, the "classical" SST relaxation method is enforced as a corrective term on the air-sea nonsolar heat flux, which is part of the model's surface boundary condition for the temperature.

The authors present an Artificial Neural Network (ANN) that mimicks this corrective heat flux term described above, and then its use and integration within the NEMO ocean model in place of the "direct" sea-surface relaxation, which requires knowledge of future SSTs, so is not possible for doing forecast. On top of it, the authors also illustrate the versatility of their methods by using their method over the 1960s, for which the observational network that would be used for constraining the model is sparse. Their results are promising and show that the ANN provides a viable alternative to classical SST relaxation, which could be used in other contexts.

The paper is well-structured, to the point and yet exhaustive. It presents a data-driven alternative to a classically used method, having the huge advantage of being predictive and not relying on an external dataset, therefore being applicable in other contexts, like forecasting and/or reanalyses over periods with poor observational coverage. The paper is quite technical, therefore making it a good candidate for GMD. I think that there are a few points on which the paper could be more accurate, and some scientific details which could be revisited. Therefore, my recommendation would be to ask the authors to take the following comments into account. Hopefully, this would make the manuscript stronger and better for the scientific community.

**Major comments:**

1) The manuscript directly introduces the method as a heat flux correction, as the title suggests. Technically, it is indeed the case, but the end goal is building a method for constraining the sea-surface temperature and reduce model biases. It is unclear whether this correction is due to fluxes being wrong – the only tangible thing is that these fluxes "suit" the model better. On a related note, the method might seem convoluted: why not simply adding an extra tendency on the SST? I think one of the main reasons, which the authors keep implicit, is that they can then directly, online integrate their ANN into the model via Fortran90 bindings. This is a strength which could be more put to the front. The method seems convoluted, but it is for a good reason, which is its integration into the classical model.

2) The variable importance scores (VIS) provide interesting insight, but it is tricky. The authors do acknowledge that air-sea interactions are strongly nonlinear, so that separating one variable from another is artificial. That said, acknowledging that the exercise is difficult to start with, and it is not possible to do a clear cut, I think that the variables that are checked against in Figure 2a) are too numerous, and some of them are too close one to another, namely:
   a. 10m wind and wind stress;
   b. Runoff, freshwater flux and latent heat – runoff is a freshwater flux, and latent heat is proportional to one component of EMP, which is essentially the freshwater flux (outside of river deltas).

3) Data-driven methods trained for RMSE reduction are known to be overtly smooth, and that problem is not easy to tackle. I would be curious to know whether the ANN also suffers from it, e.g. by seeing comparisons of instantaneous SSTs with their method vs classical relaxation.

4) Regarding section 3.2 (retrospective simulations), I think that the results are interesting but that maybe comparing them to a classical SSR run (e.g. to COBE, which is available over that period) would have been insightful. We know that the available products (like COBE) have limitations – does your method have benefits compared to directly using that?

5) There are too many commas in many sentences. I suggest the authors go over the manuscript with that in mind and reduce the number of commas.

**Minor / specific comments**

Title: is it really correcting the air-sea flux? Also please specify NEMO model version.

L. 48 – 49 : give the absolute start year instead of saying "last 15 years".

L. 51: I would add that calibrating reanalyses can be prone to double correcting some effects; i.e., deliberately introducing errors in atmosphere reanalyses that compensate systematic ones in bulk formulae.

L. 69: I think you're using a global configuration, it would be worth mentioning. Is the grid ORCA1? I'm not sure I understand what $1/3°$ - $1°$ means...

L. 72: the river discharge from land

L. 74: a TKE scheme (Gaspar et al., 1990 : https://doi.org/10.1029/JC095iC09p16179 )

L. 75: I would put the CIGAR reanalysis sentence before the model is described, in the line "We use the CIGAR parameters which is described in depth in Storto and Yang (2024); below we briefly describe some of its specific parameter choices"

L. 83: I think the word "assimilation" is misleading, since I would expect "assimilation" to refer to using an observation operator and directly assimilation measurements… maybe "corrective methodology"?

L. 93 – 95: I do not understand this sentence. What does "whereas" mean here? Where is the contradiction between the first and second parts? It would also be good to explain the readers what "unrolled" mean. I feel like the manuscript would be clearer if it first gave the predictors (the list of six variables); and then say that the predictors have information about the geographical locations. And after that, explicitly write that the SST observations are not included as predictors, which is the point of your study.

L. 103: I would say, "assuming it is nominally valid at the end of each daily window (midnight UTC)". The average itself is defined over a time interval, but your inference assumes it is valid as an instantaneous snap.

L. 103 – 105: please clearly explain what the REF model does in the presence of sea ice. "the use of the sea-ice mask in the construction of the $Q_{rp}$ field" is vague. If in the presence of sea ice, no correction is applied in REF, then say it explicitly. Does "no sea ice predictors are used" mean that the ANN is simply not used in the presence of sea ice? It would be worth clarifying.

L. 111: we progressively improve the performances…

Table 1: in my opinion, the VIS explanations belong more in the main body text than in the figure caption.

L. 131: The nonlinear character doesn't make the problem irreducible in itself. It is the **cross-variable** nonlinearities which entangle the processes that makes it complicated. I think "strong multivariate nonlinearities of air-sea interactions (e.g., wind stress depending on both near-surface winds and local temperatures via nonlinear bulk formulas)" would be more accurate.

L. 146: the sensible heat flux related to mesoscale has already been made l. 137.

L. 152: please specify that NEMO is also implemented in Fortran.

Figure 4: I think those are seasonal climatologies, right? If so, please specify.

---

## Author Comment (AC2)

**Response to Reviewers – Manuscript gmd-2024-185 "Correction of Air-Sea Heat Fluxes in the NEMO Ocean General Circulation Model Using Neural Networks" by Storto et al., 2024**

We thank the reviewers and the Editor for the careful reading and the suggestions to improve the quality and readability of the manuscript. Below, we provide a detailed reply to each of their comments (Reviewers' comment/question in bold, our reply in italic font, and new text in the manuscript in blue).

**Response to Reviewer 1**

**The use of artificial neural networks (ANNs) to model nonlinear relationships between atmospheric and oceanic state predictors supported by stationary predictors and heat flux errors appears remarkably effective.**

*Thank you for the encouraging comments.*

**It would be of interest to include in the manuscript information on the performance of the neural network in predicting the nudging increments using standard metrics like R2 or normalized mean square error or error maps. It appears that the network is able to learn a substantial part of the corrections. Figure 4 provides a hint.**

Thanks for the suggestion, we now include the normalized (and not) error map applied to the validation (independent) data. See below. The figure is indeed interesting, and shows the low residual error (<10% everywhere, on average equal to 4% = 1.36 W/m2), with the dominant effect of variability error compared to the negligible systematic error of the ANN reconstruction. We have added a paragraph in the text to comment on the new figure:

In Figure 2 we show the error maps of the inferred heat flux correction from validation (i.e., independent) data. The Normalized RMSE (panel a) shows errors smaller than 10%, and on average equal to 4% (corresponding to 1.36 W m-2); while errors peak in areas of large mesoscale activity (western boundary currents and ACC), there exist other non-obvious local peaks. The systematic error of the ANN reconstructions is very low (panel c), generally not exceeding 0.7 W m-2, indicating that the RMSE is explained, to a great extent, by random errors (panel d shows the standard deviation of the differences).

[Figure]

*Figure R1. Error maps, calculated with validation (namely independent) data, as explained in the main text. a) normalized RMSE; b) RMSE in units of heat flux (W m-2); c) bias (W m-2); d) standard deviation of differences (W m-2).*

**The method has been demonstrated to be effective in the ocean only model. Non-linear feedbacks in the coupled models could affect the effectiveness of the ANN-based corrections. Do you foresee the method to work well in the coupled models? Perhaps applicability in the context of coupled modelling/coupled reanalysis could be discussed.**

*Thanks for this suggestion. We have now included a discussion on the potential for air-sea coupled modelling, extending the sentence already mentioning the issue in the original version of the manuscript.*

The neural network-based heat flux correction method has proven effective in the ocean-only model by correcting systematic air-sea heat flux biases and improving surface and subsurface ocean temperature predictions. However, applying this method within a coupled ocean-atmosphere model may benefit climate drift correction (e.g., Gupta et al., 2013) but introduces additional complexities due to nonlinear coupled feedback. In a coupled model, the atmosphere could respond to modified fluxes in a nonlinear and potentially unpredictable manner. Heat flux corrections that work well in an uncoupled system may introduce unintended biases when the atmosphere reacts dynamically, potentially leading to unrealistic SST adjustments. Atmospheric variability (e.g., cloud cover, wind stress, and humidity) will alter in response to changes in SST, which could impact the efficacy of the NN-based correction. Corrections applied at short timescales may also have long-term impacts on coupled modes of variability (e.g., ENSO, MJO).

To make the NN approach more suitable for coupled applications, it could be retrained using data from coupled model reanalyses (e.g., CMIP simulations or CERA reanalysis datasets, for instance, Chapman and Berner, 2024). This would allow the NN to learn heat flux corrections in a system where atmospheric responses are accounted for, in analogy with flux correction or flux adjustment techniques (e.g., Sausen et al., 1988). The NN-based correction could be implemented to maintain the overall coupled energy balance while addressing systematic errors.

*Sausen, R., Barthel, K. & Hasselmann, K. Coupled ocean-atmosphere models with flux correction. Climate Dynamics 2, 145–163 (1988). https://doi.org/10.1007/BF01053472*

*Gupta, A. S., N. C. Jourdain, J. N. Brown, and D. Monselesan, 2013: Climate Drift in the CMIP5 Models. J. Climate, 26, 8597–8615, https://doi.org/10.1175/JCLI-D-12-00521.1.*

*Chapman, W. E., and Berner, J.. A State-Dependent Model-Error Representation for Online Climate Model Bias Correction. ESS Open Archive . November 23, 2024. doi:10.22541/essoar.172526800.05354621/v2*

**Have you tested other ANN architectures like CNN? While CNN may be more difficult to apply online, it would be interesting to assess if it would be superior to the simple column model?**

*Unfortunately, it is difficult to test architectures embedding convolutional layers within our framework; this is mostly because of technical challenges linked with online inference in NEMO. Indeed, convolutional layers need to know exactly the MPI decomposition of NEMO, because each iteration requires data exchange across neighboring domains/cores. Inference software that is available at the moment (our ANNIF module, ECMWF/Infero, neural-fortran, or Cambridge-ICCS/fortran-tf-lib) does not allow specifying an MPI decomposition of the calling module. Another solution is being developed, based on leveraging Python API for the OASIS coupler (see Barge and Le Sommer, 2024), but it will be released in the future for the latest NEMO version, so for the time being we were forced to choose a model architecture not relying on convolutional layers. We have added a short discussion on that, in the final section:*

In this study, we demonstrate the significant impact of online inference, which allows high-frequency (3-hourly) updates of the correcting fields. For this reason, testing different model architectures, e.g., those

relying on convolutional layers, which require MPI communication across NEMO domains inside the convolutional filters, was technically complex and demanding. It is not obvious whether convolutional layers are beneficial compared to grid-point-wise corrections (see, e.g., different conclusions in Chen et al., 2022; Chapman and Berner, 2024), as the potential advantage of retaining horizontal patterns is balanced by the computational needs of coarsening the spatial resolution. In the future, more sophisticated inference libraries and tools for online prediction are expected to be available, paving the way for testing different neural network architectures.

**References**

Barge, A. and Le Sommer, J.: Online deployment of pre-trained machine learning components within Earth System models via OASIS, EGU General Assembly 2024, Vienna, Austria, 14–19 Apr 2024, EGU24-16148, https://doi.org/10.5194/egusphere-egu24-16148, 2024.

---

## Author Comment (AC3)

**Response to Reviewers – Manuscript gmd-2024-185 "Correction of Air-Sea Heat Fluxes in the NEMO Ocean General Circulation Model Using Neural Networks" by Storto et al., 2024**

We thank the reviewers and the Editor for the careful reading and the suggestions to improve the quality and readability of the manuscript. Below, we provide a detailed reply to each of their comments (Reviewers' comment/question in bold, our reply in italic font, and new text in the manuscript in blue).

**Response to Reviewer 2**

**The paper is well-structured, to the point, and yet exhaustive. It presents a data-driven alternative to a classically used method, having the huge advantage of being predictive and not relying on an external dataset, therefore being applicable in other contexts, like forecasting and/or reanalyses over periods with poor observational coverage. The paper is quite technical, therefore making it a good candidate for GMD. I think that there are a few points on which the paper could be more accurate, and some scientific details which could be revisited. Therefore, my recommendation would be to ask the authors to take the following comments into account. Hopefully, this would make the manuscript stronger and better for the scientific community.**

Thank you very much for the positive and encouraging comments, and for all the general and detailed corrections that will improve the manuscript.

Major comments:

**1) The manuscript directly introduces the method as a heat flux correction, as the title suggests. Technically, it is indeed the case, but the end goal is building a method for constraining the sea-surface temperature and reduce model biases. It is unclear whether this correction is due to fluxes being wrong – the only tangible thing is that these fluxes "suit" the model better. On a related note, the method might seem convoluted: why not simply adding an extra tendency on the SST? I think one of the main reasons, which the authors keep implicit, is that they can then directly, online integrate their ANN into the model via Fortran90 bindings. This is a strength which could be more put to the front. The method seems convoluted, but it is for a good reason, which is its integration into the classical model.**

Thank you for this comment. The method, while acting on the heat flux, accounts indeed for all possible sources of errors, including e.g. vertical mixing, etc. In practice, it is an SST bias correction, where we assess the bias against the observations and cannot attribute the bias source unambiguously. We follow the Reviewer's recommendation and change the title to **"Correction of Sea Surface Biases in the NEMO Ocean General Circulation Model Using Neural Networks"** and discuss this in more detail in the text. The correction is not added to the SST tendency because this is in our experience never satisfying; either one should arbitrarily choose a vertical propagation strategy (e.g., within the mixed layer?), or if applied only to the surface it may disappear throughout the exchange with the atmosphere. It is therefore customary in NEMO to correct the air-sea heat fluxes. We have added this paragraph and adjusted some parts of the abstract and introduction accordingly.

The proposed methodology, while formulated as a correction to air-sea heat fluxes, effectively accounts for multiple sources of bias in the modeled sea surface temperature (SST), including potential errors in vertical mixing and other oceanic processes. Since the bias is assessed against observations without possible attribution to a specific error source, the method serves as a general SST bias correction strategy. Additionally, the correction is applied to the air-sea heat flux rather than directly modifying the SST tendency. Direct SST tendency corrections are generally unsatisfactory, as they require arbitrary assumptions about vertical propagation - such as confinement within the mixed layer - or risk being nullified by air-sea interactions (see, e.g., Waters et al., 2015; Storto and Oddo, 2019). Adjusting air-sea heat fluxes is

therefore a customary and physically consistent practice in ocean general circulation models; similar approaches are indeed used also by state estimation systems, such as ECCO4 (Forget et al., 2015), which employs observations to correct heat flux components.

*Waters, J., Lea, D.J., Martin, M.J., Mirouze, I., Weaver, A. and While, J. (2015), Implementing a variational data assimilation system in an operational 1/4 degree global ocean model. Q.J.R. Meteorol. Soc., 141: 333-349. https://biblioproxy.cnr.it:2481/10.1002/qj.2388*

*Storto A, Oddo P. Optimal Assimilation of Daytime SST Retrievals from SEVIRI in a Regional Ocean Prediction System. Remote Sensing. 2019; 11(23):2776. https://doi.org/10.3390/rs11232776*

*Forget G, Campin J-M, Heimbach P, Hill CN, Ponte RM, Wunsch C (2015) ECCO version 4: an integrated framework for non-linear inverse modeling and global ocean state estimation. Geosci Model Dev 8:3071–3104. https://doi.org/10.5194/gmd-8-3071-2015*

**2) The variable importance scores (VIS) provide interesting insight, but it is tricky. The authors do acknowledge that air-sea interactions are strongly nonlinear, so that separating one variable from another is artificial. That said, acknowledging that the exercise is difficult to start with, and it is not possible to do a clear cut, I think that the variables that are checked against in Figure 2a) are too numerous, and some of them are too close one to another, namely:**

**a. 10m wind and wind stress;      b. Runoff, freshwater flux and latent heat – runoff is a freshwater flux, and latent heat is proportional to one component of EMP, which is essentially the freshwater flux (outside of river deltas).**

We have added some comments on that, stressing even more that separating one variable from another is artificial. Indeed, while there are strong correspondences, the use of wind versus wind stress allows rectifying the impact of ocean surface currents in the bulk formulas, and so on for the other predictors; we thus believe that VIS provides only a general qualitative insight, and stressed this comment in the text.

**3) Data-driven methods trained for RMSE reduction are known to be overtly smooth, and that problem is not easy to tackle. I would be curious to know whether the ANN also suffers from it, e.g. by seeing comparisons of instantaneous SSTs with their method vs classical relaxation.**

Thanks. The grid-point-wise correction implies that the smoothness of the ANN-based correction depends on that of the predictors, i.e. the model fields. We verified some snapshots (see the example below for a random realization in summertime) and found high consistency between the original output and the NN-inferred one. We prefer not to add another figure on the "instantaneous reconstruction" but mention this in the revised version anyway, and include the map of RMSE as requested by Reviewer 1.

[Figure]

*Figure R2. Comparison of instantaneous air-sea heat flux reconstruction made by the ANN (right panel) with the corresponding correction obtained with SST nudging.*

**4) Regarding section 3.2 (retrospective simulations), I think that the results are interesting but that maybe comparing them to a classical SSR run (e.g. to COBE, which is available over that period) would have been insightful. We know that the available products (like COBE) have limitations – does your method have benefits compared to directly using that?**

Thank you for this comment. We have experimented with nudging to COBE SST (experiment FB01), and the results show a positive impact of the nudging scheme, although it is generally smaller (See below). We will consider whether to include the panel or mention it only within the text.

[Figure]

*Figure R3. Comparison of temperature RMSE decrease between the experiment with COBE SST nudging (FB01) and NN-based correction (NNC) Over the period 1961-1979.*

**5) There are too many commas in many sentences. I suggest the authors go over the manuscript with that in mind and reduce the number of commas.**

We will revise the manuscript keeping in mind this comment, thank you.

**Minor / specific comments**

**Title: is it really correcting the air-sea flux? Also please specify NEMO model version.**

We changed it according, thank you (see the answer to the first major point). We prefer not to mention the version of NEMO in the title but we added it in the abstract, as our method is modular and technically ready to use for any previous and future version of NEMO.

**L. 48 – 49 : give the absolute start year instead of saying "last 15 years".**

Changed

**L. 51: I would add that calibrating reanalyses can be prone to double correcting some effects; i.e., deliberately introducing errors in atmosphere reanalyses that compensate systematic ones in bulk formulae.**

We prefer to keep the original formulation here. Indeed, the works we cite (JRA55-do, COREv2, etc.) correct the input atmospheric reanalysis (t2m, downwelling radiation, etc.) based on comparison with observational datasets. This means that errors from bulk are not back-propagated to atmospheric reanalysis input, in principle; on the contrary, errors in the bulk formulas are not corrected (except for global constraints).

**L. 69: I think you're using a global configuration, it would be worth mentioning. Is the grid ORCA1? I'm not sure I understand what 1/3º - 1º means…**

Changed according to the reviewer's suggestion.

**L. 72: the river discharge from land**

Corrected

**L. 74: a TKE scheme (Gaspar et al., 1990 : https://doi.org/10.1029/JC095iC09p16179 )**

Corrected

**L. 75: I would put the CIGAR reanalysis sentence before the model is described, in the line "We use the CIGAR parameters which is described in depth in Storto and Yang (2024); below we briefly describe some of its specific parameter choices"**

Corrected

**L. 83: I think the word "assimilation" is misleading, since I would expect "assimilation" to refer to using an observation operator and directly assimilation measurements... maybe "corrective methodology"?**

Thanks for the comment. However, we prefer to keep the sentence as it is, as we believe that nudging/Newtonian relaxation schemes are proper data assimilation schemes, although simpler than others. This is also consistent with similar literature.

**L. 93 – 95: I do not understand this sentence. What does "whereas" mean here? Where is the contradiction between the first and second parts? It would also be good to explain the readers what "unrolled" mean. I feel like the manuscript would be clearer if it first gave the predictors (the list of six variables); and then say that the predictors have information about the geographical locations. And after that, explicitly write that the SST observations are not included as predictors, which is the point of your study.**

We have reformulated the sentence to make it clear, thanks. And also added the reviewer's comment.

**L. 103: I would say, "assuming it is nominally valid at the end of each daily window (midnight UTC)". The average itself is defined over a time interval, but your inference assumes it is valid as an instantaneous snap.**

Corrected

**L. 103 – 105: please clearly explain what the REF model does in the presence of sea ice. "the use of the sea-ice mask in the construction of the Qrp field" is vague. If in the presence of sea ice, no correction is applied in REF, then say it explicitly. Does "no sea ice predictors are used" mean that the ANN is simply not used in the presence of sea ice? It would be worth clarifying.**

Clarified. We indeed use the sea-ice mask in the nudging experiment used for training. See our relaxation routine, line 122 at https://baltig.cnr.it/nemo_ismar-rm/nemo_4.0.7/-/blob/main/cfgs/MY_SRC/sbcssr.F90 , the nudging-based correction is multiplied by *exp ( -fr_i(ji,jj)\*fr_i(ji,jj)/ 0.16_wp )* , meaning that for value of SIC larger than zero an exponential decay function is used to attenuate the correction, which is the same correction learned by the neural networks. Additionally, the lack of sea-ice predictors and the fact that SST observations are extrapolated to be close to the freezing temperature beneath sea ice makes the methodology not suitable for correcting fluxes in the presence of sea ice.

Over sea-ice-covered areas, the heat flux corrections vanish, due to the use of a sea-ice-based weighting function - that zeros the correction for non-zero values of the sea-ice concentration - in the construction of the $Qrp$ fields in the nudging experiment. The nudging experiment is also used in the training of the ANN, thus resulting in negligible corrections therein. Additionally, no sea-ice predictors are used. This is because sea surface temperature data beneath sea ice are extrapolated from sea ice concentration data and are less reliable (Rayner et al., 2003).

**L. 111: we progressively improve the performances…**

Corrected

**Table 1: in my opinion, the VIS explanations belong more in the main body text than in the figure caption.**

Corrected

**L. 131: The nonlinear character doesn't make the problem irreducible in itself. It is the cross-variable nonlinearities that entangle the processes that makes it complicated.**

Corrected

**I think "strong multivariate nonlinearities of air-sea interactions (e.g., wind stress depending on both near-surface winds and local temperatures via nonlinear bulk formulas)" would be more accurate.**

Corrected

**L. 146: the sensible heat flux related to mesoscale has already been made l. 137.**

Reformulated

**L. 152: please specify that NEMO is also implemented in Fortran.**

Added

**Figure 4: I think those are seasonal climatologies, right? If so, please specify.**

Clarified